# Unraveling the Interplay of KRAS, NRAS, BRAF, and Micro-Satellite Instability in Non-Metastatic Colon Cancer: A Systematic Review

**DOI:** 10.3390/diagnostics14101001

**Published:** 2024-05-12

**Authors:** Elena Orlandi, Mario Giuffrida, Serena Trubini, Enrico Luzietti, Massimo Ambroggi, Elisa Anselmi, Patrizio Capelli, Andrea Romboli

**Affiliations:** 1Department of Oncology-Hematology, Piacenza General Hospital, 29121 Piacenza, Italy; s.trubini@ausl.pc.it (S.T.); m.ambroggi@ausl.pc.it (M.A.); e.anselmi@ausl.pc.it (E.A.); 2Department of General Surgery, Piacenza General Hospital, 29121 Piacenza, Italy; m.giuffrida@ausl.pc.it (M.G.); e.luzietti@ausl.pc.it (E.L.); p.capelli@ausl.pc.it (P.C.); a.romboli@ausl.pc.it (A.R.)

**Keywords:** colonic neoplasms, microsatellite instability, mutation, neoplasm staging, prognosis proto-oncogene proteins B-raf, proto-oncogene proteins p21(ras)

## Abstract

Microsatellite Instability (MSI-H) occurs in approximately 15% of non-metastatic colon cancers, influencing patient outcomes positively compared to microsatellite stable (MSS) cancers. This systematic review focuses on the prognostic significance of KRAS, NRAS, and BRAF mutations within MSI-H colon cancer. Through comprehensive searches in databases like MEDLINE, EMBASE, and others until 1 January 2024, we selected 8 pertinent studies from an initial pool of 1918. These studies, encompassing nine trials and five observational studies involving 13,273 patients, provided insights into disease-free survival (DFS), survival after recurrence, and overall survival. The pooled data suggest that while KRAS and BRAF mutations typically predict poorer outcomes in MSS colorectal cancer, their impact is less pronounced in MSI contexts, with implications varying across different stages of cancer and treatment responses. In particular, adverse effects of these mutations manifest significantly upon recurrence rather than affecting immediate DFS. Our findings confirm the complex interplay between genetic mutations and MSI status, emphasizing the nuanced role of MSI in modifying the prognostic implications of KRAS, NRAS, and BRAF mutations in colon cancer. This review underscores the importance of considering MSI alongside mutational status in the clinical decision-making process, aiming to tailor therapeutic strategies more effectively for colon cancer patients.

## 1. Introduction

Colorectal cancer (CRC) is the third leading cause of cancer deaths in the world with a rate of 36.6 new cases per 100,000 men and women per year [1,2]. Localized CRC accounts for 35.5% of CRC diagnoses, with a five-year relative survival of 90.9%, while the regional lymph node spread (stage III) occurs in 36% of cases, and the five-year relative survival is 73.4% [2]. Approximately 15% of colon cancer cases are characterized by the incompetence of the DNA mismatch repair (MMR) system: MMR proteins (MLH1, MSH2, MSH6, and PMS2) are nuclear enzymes that bind to areas of abnormal DNA and repair base–base mismatch during cellular proliferation and division [3]. Defects in DNA MMR genes can lead to the insertion or deletion of repeating nucleotide sequences in a process known as Microsatellite Instability (MSI), leading to abnormal shortening or lengthening of repeating base pair units of DNA [4]. MSI is largely due to MLH1 inactivation through hypermethylation of the promoter in sporadic CRC; on the other hand, in Lynch syndrome, MSI is mostly due to an inherited germline mutation of the MMR gene [5]. MSI is more frequent in stage II (almost 20%) and III (12%) tumors than in stage IV tumors (4%) [6].

A recent study conducted in the USA showed that the expression and mutation patterns of mismatch repair proteins like MLH1, MSH2, MSH6, and PMS2 are significantly correlated with MSI status and survival rates in non-metastatic colorectal cancer, suggesting a critical role for genetic screening in therapeutic decision-making [3].

International research, including systematic reviews and European studies, has demonstrated that MSI-high tumors generally exhibit a better prognosis and respond differently to conventional chemotherapy, supporting the use of MSI status as a crucial stratification factor in clinical trials and treatment planning [7,8].

The global and regional findings highlight the importance of adapting treatment protocols based on MSI status to optimize treatment efficacy, particularly concerning the use of immunotherapy in MSI-high patients in various international settings [7,8].

Several retrospective studies [4,9,10], a meta-analysis [7], and some large trials [6,7,8,11,12,13] support the notion that patients with MSI tumors evolve more favorably than those with microsatellite stability, but the difference in prognosis is larger for stage II than for stage III patients. In addition, retrospective studies of stages II and III colon cancer patients, analyzing data from Randomized Controlled Trials (RCTs) on adjuvant therapy, showed that stage II colon cancer patients with high microsatellite instability/deficient mismatch repair (dMMR/MSI-H) status do not benefit from adjuvant 5-FU-based chemotherapy [12,13,14,15]. For stage II, current guidelines recommend adjuvant chemotherapy treatment by stratifying patients on the basis of both clinical (comorbidities, reduced life expectancy), biological (dihydropyrimidine dehydrogenase -D PD- deficiency), and risk factors based on neoplastic disease characteristics [16]. For stage III, the indication for chemotherapy has to take into consideration a number of factors, e.g., Eastern Cooperative Oncology Group (ECOG) performance status >2, uncontrolled infection, severe liver and renal dysfunction, and heart failure [16]. Despite adjuvant chemotherapy being associated with improved Disease-Free Survival (DFS) in proficient mismatch repair (pMMR) patients, this is not confirmed in MSI patients [17,18,19]. Relapse-Free Survival (RFS) was better for patients with MSI than for microsatellite stable (MSS) CRC, regardless of the side. Overall survival (OS) was statistically significantly different between MSI and MSS CRC for right colon cancer, whereas it was not for left colon cancer [20]. KRAS mutations involving either codon 12 or 13 can be identified in 40% of tumors; they were independently associated with a worse prognosis [21,22,23,24,25,26,27], especially G12V [22], which is particularly related to an adverse outcome; also KRAS G12C and G13D were linked with rather poor survival in some studies [25]. Among MSI-H tumors, in which most of the BRAF mutations occur, the presence of a mutation does not have the same adverse prognostic significance [27,28]. Further classification of mutated CRC (including mutated BRAF, KRAS, and NRAS) might be seen in the differentiation of left-sided versus right-sided primary tumor location, probably being a surrogate for molecular profiles that have not been understood in their full extent yet [29]. BRAF V600E and KRAS mutations were significantly associated with shorter DFS and OS in patients with microsatellite-stable tumors, but they were not in patients with MSI tumors [27]. Immunotherapy has been achieving promising results in the neoadjuvant setting for those patients with the presence of MSI in both monotherapy and combination regimens, confirming the predictive role of MSI even in non-metastatic disease [30,31,32,33]. A meta-analysis showed a statistically significant overall negative effect of both KRAS and BRAF mutations on prognosis in the non-metastatic setting, with increased significance when the estimates were adjusted for the presence of MSI-H. Therefore, in light of the data about MSI-, KRAS-, NRAS-, or BRAF-mutated patients, we decided to clarify both the prognostic and predictive role of the mutations in this setting of patients and in patients undergoing adjuvant treatment [34]. 

This systematic review aims to investigate how the presence of NRAS, KRAS, or BRAF mutation in non-metastatic MSI colon cancer patients affects outcomes such as DFS, OS, and survival after recurrence (SAR).

Our goal is to enrich the existing scientific literature by clarifying these associations, thereby aiding in the development of more precise and effective therapeutic strategies tailored to this specific patient demographic.

## 2. Materials and Methods

### 2.1. Protocol and Registration

The research project was registered in the International Prospective Register of Systematic Reviews (PROSPERO) with the protocol number CRD42023495745. The report was written according to the PRISMA (Preferred Reporting Items for Systematic Review and Meta-Analysis) 2020 checklist.

### 2.2. Eligibility Criteria and Research Question

A systematic review of the presence of NRAS, KRAS, or BRAF mutation in non-metastatic MSI CRC, which included both patients undergoing and not undergoing adjuvant treatment, according to PICO criteria (PECO variant), was performed. The included population consists of patients with a diagnosis of non-metastatic MSI CRC with the presence of NRAS, KRAS, or BRAF mutation as exposure and with various survival and recurrence items as outcomes. Studies reporting any period of follow-up could be susceptible to inclusion.

We included observational prospective and retrospective cohort studies, RCTs, ongoing trials reporting original data, written in English, published in full-text format; studies about patients with MSI CRC who had undergone surgical resection, with or without following adjuvant treatment, with available BRAF, NRAS, and KRAS status; survival outcomes must be reported. There was no restriction on the time of publication. Case reports, case series, preclinical studies, and animal studies were excluded; studies involving metastatic setting, and subsequent publications with the same patients were excluded. Moreover, if the full data of abstract or original research were not recovered, as well as if primary or secondary outcomes were not reported or available after request to the corresponding author of the publication, the study was not included and reported in the Appendix A).

### 2.3. Search Strategy

We have systematically searched on MEDLINE, EMBASE, Cochrane Database of Systematic Reviews, Cochrane Central Register of Controlled Trials, and IRSCTN Registry up to 1 January 2024. The reference lists of included articles and relevant reviews were examined to identify additional relevant publications for inclusion. Furthermore, we have sought clinicaltrials.gov for any ongoing or unpublished trials and for additional info from published data.

A search string in MEDLINE was performed and included relevant mesh in the research field mixed with Boolean operators “AND” and “OR”. The following syntax was used for the search: (“Microsatellite Instability”[Mesh] AND ((“KRAS protein, human” [Supplementary Concept] OR “Genes, ras”[Mesh] OR “BRAF protein, human” [Supplementary Concept]) AND “Colorectal Neoplasms”[Mesh])) NOT (neoplasm metastasis[MeSH Terms]).

The search string used in EMBASE was the following: Colorectal neoplasm AND NOT Metastatic and microsatellite instability AND RAS Mutation AND BRAF.

### 2.4. Data Extraction

Selected articles from each search string underwent a duplicate identification; duplicates were removed with the consecutive use of two programs: Systematic Review Accelerator for the first step deduplication and Rayyan for the second step. Subsequently, two reviewers (A.R. and M.G.) independently screened titles and abstracts of all references identified from the initial search. Any disagreements were resolved by a third reviewer (E.O.). Full-text articles of potentially relevant publications were scrutinized in detail, and inclusion criteria were applied to select eligible articles. The exclusion criteria will be documented in Appendix A. Agreement was recorded at each stage; disagreements between reviewers were resolved through consensus or by discussion with a third independent reviewer (E.O.). Rayyan was used to support citation screening, full-text review, and export of data and references.

From each eligible study, two reviewers independently extracted relevant information, using a pre-specified standardized extraction form. Any disagreement between reviewers will be resolved as outlined above. Data from included studies were extracted for study characteristics such as the first author, year of publication, sample size (total population); participant demographics such as MSI (% total), sex, stage of disease with relative percentage of stage I, II, and III tumors, details of adjuvant treatment, percentage of KRAS, NRAS, and BRAF mutations, and outcomes (overall and by subgroup), type of Cox-regression model for hazard ratio (HR) estimation (univariate vs. multivariate), and covariates included and/or adjusted for in case of multivariable models. All data were extracted using a shared and pre-established form and then transferred into an Excel spreadsheet. Extracted data elements also included outcome measures such as OS, DFS, and SAR, the size of the association (OR, RR, or HR) with corresponding 95% CI and factors adjusted for, confounding factors taken into consideration, and methods used to control covariates.

The synthesis of different items of interest among included studies are reported in different groups that quote DFS, OS, and SAR according to the presence of KRAS, NRAS, or BRAF mutation in both MSI and MSS populations.

### 2.5. Primary and Secondary Outcomes

The primary outcome was DFS, defined as the measure of time after treatment during which no sign of cancer is found. The secondary outcomes we investigated were OS, defined as the time from treatment to death, regardless of disease recurrence, and SAR, defined as the duration between the detection of the initial recurrence and either death or the last follow-up.

### 2.6. Quality Assessment

The methodological quality and potential risk of bias of included studies were assessed at the outcome level independently by two reviewers using the Newcastle-Ottawa scale, a validated tool to evaluate the quality of nonrandomized studies (Appendix A) [35].

## 3. Results

The initial search retrieved 1918 articles, which were reviewed for entry criteria. After excluding 1788 records, 126 papers were assessed for eligibility; after two rounds of review, according to the inclusion and exclusion criteria, eight articles were detected (Figure 1).

We identified one pooled analysis including seven trials (IDEA France [NCT00958737], Alliance [NCCTG-N0147], NSABP-C07 [NCT00004931], and NSABP-C08 [NCT00096278], CALGB 89803 [NCT00003835], PETACC3 [NCT00026273], PETACC8 [NCT00265811]) [36], a mixed pooled analysis from the QUASAR 2 clinical trial, and an Australian community-based series [37], the trial QUASAR [38], and five observational studies that met the inclusion criteria [39,40,41,42,43]. The population characteristics of the studies, the disease stage, the percentage of KRAS and BRAF mutations in MSI patients, and the types of adjuvant treatments are outlined in Table 1.

The reasons for the report that might appear to meet inclusion criteria and were not included in the review are reported in Appendix A.

### 3.1. Effect of Mutations in DFS, OS, and SAR

The data regarding the investigated survival outcomes are described in Table 2.

Regarding DFS, only one study identifies a higher relative risk (RR) tendency in BRAF-mutated MSI compared to BRAF wild type, with an RR of 1.32 (CI 0.8–2.16) [38]; no statistically significant difference was reported by Taieb and Kadowaki [36,42]. While Domingo [37] did not report a significant increase in DFS in the MSI patient group, a significant HR was evident for both BRAF- and KRAS-mutated MSI, when compared to the wild-type, MSS population.

Statistical significance in OS was reported by Batur [40] concerning BRAF-mutated patients versus wild type with *p* = 0.01. However, Maestro and Taieb did not identify a statistically significant difference [36,39]; the overall MSI group, irrespectively of mutational status, exhibited better OS compared to MSS patients (HR 0.33 and 0.67, respectively).

Taieb’s pooled analysis is the only one that studies SAR, highlighting a statistically significant difference in adverse prognostic terms for those MSI patients with BRAF and KRAS mutations in the multivariate analysis.

### 3.2. Pooled Analysis and RCTs

The pooled analysis of seven trials within the ACCENT/IDEA consortium [36] explores the distinct prognostic implications of KRAS exon 2 submutations and the BRAF V600E mutation in stage III colon cancer. Patients with stage I, II, and IV colon cancer, or those with lower or middle rectal cancer, or individuals who did not undergo chemotherapy were excluded. The study focuses on the differentiation between 7492 MSS and 968 MSI cases, all patients receiving adjuvant chemotherapy. Among the 968 MSI-H tumors, 18.1% were KRAS mutants and 40.6% were BRAF V600E mutants. The findings elucidate that both mutations in the entire population were associated with shorter Time to Recurrence (TTR), OS, and SAR compared to patients without these mutations, confirmed by multivariate analysis. An interaction test between the mutational status and treatment suggests that the negative prognostic impact of KRAS and BRAF V600E mutations on TTR and OS could be assessed independently by the treatment administered. In the MSI-H cohort, TTR showed no difference between KRAS MT or BRAF MT tumors and DWT tumors. Considering OS, no significant impact was observed in KRAS MT patients, while BRAF MT patients had shorter OS than DWT patients (*p* = 0.042). SAR was shorter in KRAS and BRAF MT patients (*p* < 0.001). No association was found between KRAS and BRAF V600E mutational status and TTR or OS in the multivariable analysis for MSI-H patients. However, SAR was worse in both KRAS MT (*p* = 0.017) and BRAF MT subgroups (*p* < 0.01) compared to the DWT cohort, suggesting that these mutations are poor prognostic markers after disease recurrence, regardless of MSI status. In particular, the RCT of Ogino [44], included in the pooled analysis of Taieb, investigated the predictive significance of the combined status of BRAF and MSI in relation to adjuvant treatment: the findings indicated that there was no noticeable advantage of IFL over FU/LV in either BRAF-mutated MSI-H or BRAF wild-type MSS patients. In Gavin’s [45] analysis of 201 stage II and III MSI patients, no significant interactions were observed between mutations (including BRAF, KRAS, NRAS) or MMR status and oxaliplatin treatment. These mutations and MMR status do not confer resistance to the beneficial effects of oxaliplatin in tumors. 

The mixed analysis of Domingo [37] utilizes data from the QUASAR 2 clinical trial and an Australian community-based series in stage II or III CRC, to investigate the impact of mutation burden and other molecular factors on the prognosis of CRC patients undergoing curative therapies. Patients enrolled in the QUASAR 2 trial were characterized as having high-risk stage II or stage III CRC. These patients were randomly assigned to receive either capecitabine alone or a combination of capecitabine and bevacizumab, without the inclusion of radiotherapy. The community-based series comprised patients diagnosed with stage II or III CRC cancer who underwent standard neoadjuvant or adjuvant fluorouracil-based chemotherapy or concurrent chemoradiotherapy. The investigation examined various combinations of MSI (244 patients), KRAS mutation, and BRAF mutation: in contrast to the triple-negative group (MSI-negative, KRAS, and BRAF wild type), MSI-positive CRC with KRAS (*p* = 0.028) or BRAF (*p* = 0.017) mutations exhibited a notably improved prognosis. However, this difference did not reach statistical significance when compared to MSI-positive cancers without KRAS or BRAF mutations.

The QUASAR [38] analysis investigates the predictive value of MMR, KRAS, and BRAF mutations in CRC for recurrence and the potential benefits from chemotherapy. Subjects were randomly assigned to receive FU/FA chemotherapy (*n* = 1622) or undergo observation only (*n* = 1617), with the option of introducing chemotherapy in case of a relapse. The recurrence risk for dMMR tumors was approximately half that of pMMR tumors (RR 0.53; *p* < 0.001). Although the risk of recurrence was significantly higher for KRAS mutant compared to wild-type tumors, in BRAF-mutated and KRAS-mutated patients, the RR data consistently showed a benefit for MSI patients (RR 0.57, *p* = 0.001, and RR 0.52, *p* = 0.0004, respectively).

### 3.3. Observational Studies

In the observational study conducted by De Cuba [41], 143 samples from patients diagnosed with stage II and III MSI colon cancers between 1987 and 2008 were evaluated. The five-year Cancer-Specific Survival was significantly worse in cases with mutated BRAF or KRAS, with a *p*-value of 0.04, and this significance persisted in the multivariate analysis. While the mutation status versus the wild type did not show significant prognostic value for OS, there was a trend toward worse survival when mutations in these genes were present. 

The analyses by Batur [40] and Maestro [39] present conflicting results, likely due to their sample sizes being too small to yield statistically significant outcomes. Specifically, the Batur analysis confirmed progress toward a worse prognosis for patients with BRAF-mutated plus MSI tumors, compared to those with BRAF-negative plus MSI tumors (*p* = 0.001). On the other hand, the OS analysis of MSI stratified by BRAF in 351 CRC patients conducted by Maestro did not reveal statistically significant differences.

Two observational studies [42,43] aimed to evaluate the prognostic implications of these molecular markers in the Japanese population, considering that the prevalence of BRAF mutations and MSI-H in the Asian population was lower than observed in Western populations. In the cohort of Kadowaki S et al. [42], despite the sample limitations, KRAS or BRAF mutations were linked to poorer survival, regardless of MSI status. However, in the study by Nakaji [43], which investigated the prognostic significance of the BRAF V600E mutation and MSI in sporadic CRC, there were no survival differences in the MSI-H group between the BRAF V600E mutation and BRAF wild-type groups (*p* = 0.4655).

The survival data from the included studies in Table 2 are reported as a forest plot in Figure 2 to assess the distribution of hazard ratios for each mutation across different reported survival outcomes.

## 4. Discussion

### 4.1. MSI Research and Clinical Implications

A growing trend in the study of MSI in colon cancer patients has become evident. To illustrate, there was a substantial increase in publications on PubMed from 2006 to 2023. In 2006, a total of 72 articles were published, whereas, by 2023, this figure had surged to approximately four times that amount, reaching 275 (Figure 3).

The assessment of MSI status has emerged as a valuable prognostic tool in CRC. Tumors exhibiting MSI often demonstrate distinctive clinicopathological features and a more favorable prognosis compared to MSS tumors. The identification of MSI status is then crucial for therapeutic decision-making.

Among patients with localized colon cancer, MSI/dMMR status defines a subgroup of patients with a good prognosis but a less expected benefit from chemotherapy [12,46,47]. In particular, MSI may be useful to identify a small (10–15%) subset of stage II patients who have a very low risk of recurrence, in whom the benefits of fluoropyrimidines have not been demonstrated yet, and thus adjuvant chemotherapy should not be indicated [13].

Trials with immunotherapy in combination with chemotherapy regimens in the adjuvant setting have been still ongoing, in particular for stage III colon cancer with MSI, extended to patients with POLE exonuclease domain mutation, confirming the peculiar role of this specific subgroup of patients and the need to further investigate their clinical characteristics and response to therapy [48,49].

### 4.2. BRAF and RAS Mutations: Comparative Analysis with Other Studies

BRAF and RAS mutations, especially KRAS and NRAS alterations, represent significant molecular events in CRC pathogenesis [3]. In particular, CRC NRAS mutations seem to be a different molecular subset, enriched in left-sided primary tumors and among African Americans, associated with a poorer prognosis and worse outcomes than either KRAS-mutant or wild-type CRC [50]. Tumors with mutated KRAS and NRAS are unresponsive to anti-EGFR therapy because mutations within proteins located downstream of the EGFR lead to constitutive activation of the pathway, even if the EGFR is blocked. Therefore, these mutations are a negative predictive factor for a biological therapy response [51]. BRAF activating mutations, most of which occur in codon 600 (V600E), happen in less than 10% of sporadic CRCs and are a strong negative prognostic marker for both early-stage and advanced/recurrent non-MSI tumors [10,18,25,26,52,53,54]. Beyond their prognostic role, RAS and BRAF mutations exhibit reduced responsiveness to standard chemotherapeutic regimens, as these alterations confer resistance mechanisms that compromise the efficacy of therapeutic interventions [22,24].

Unlike the straightforward prognostic implications in MSS tumors, the interaction between these mutations and MSI presents a nuanced challenge. Some studies have shown a negative prognostic impact of BRAF and KRAS mutations in MSI settings, but our findings suggest that these mutations do not uniformly predict poor outcomes across all MSI tumors.

While MSI-H tumors are generally associated with a better prognosis, the presence of concurrent RAS or BRAF mutations within these MSI-H cases could exacerbate the negative impact on patient outcomes.

The recent meta-analysis by Formica et al. [34] suggests that tumors with mutations in BRAF/KRAS in the MSS context exhibit a more aggressive tumor biology, which is evident from the early stages of cancer growth and persisting into the metastatic stage. This finding was confirmed with the estimates adjusted for the presence of MSI-H status, resulting in that effect being notably pronounced.

Our analysis underscores that, despite the robust prognostic impact of MSI, the presence of KRAS and BRAF mutations seems to exert a detrimental influence as the disease advances, outweighing the potential favorable effects conferred by MSI. The most recent pooled analysis by Taieb highlights how the divergent prognostic impact of mutations in this population is confined to SAR but not short-term survival outcomes such as RFS. The known correlation between MSI and immunity contrasts with a recently confirmed opposing role of KRAS and BRAF mutations in the immune response in CRC. KRAS-mutated tumors exhibit decreased immune cell infiltration compared to KRAS wild-type tumors, whereas BRAF-mutated tumors, particularly concerning cytotoxic T cells and Th1 cells, demonstrate the opposite effect [55]. BRAF-mutated tumors have been associated not only with increased immune cell infiltration but also with the expression of immunotherapeutic targets [56]; however, the specific contributions of mutated BRAF and MSI to the immune response remain uncertain. Studies on immunotherapy in metastatic patients have revealed a potential benefit in the use of immune checkpoint inhibitors in MSI patients with BRAF mutations, despite the prognostic value of such mutations [55]. It is crucial to consider that most analyses assessed by clinical trials and observational studies belong to an era when the potential for immunotherapy has not been yet available for patients with recurrent metastatic tumors, especially those with BRAF-mutated MSI. Therefore, this constitutes a significant limiting factor.

### 4.3. Strength and Limitations

This systematic review synthesizes a comprehensive range of studies examining the prognostic significance of NRAS, KRAS, and BRAF mutations within non-metastatic MSI colon cancer. A major strength of this review is its extensive coverage of diverse genetic profiles, which provides a nuanced understanding of how these mutations influence patient outcomes across different populations. The review utilizes rigorous methodological standards to ensure that the findings are reliable and replicable, enhancing the utility of the conclusions drawn for clinical practice.

Despite its strengths, this review has several limitations. The studies included may have varied in design and scope, potentially introducing heterogeneity that could affect the interpretation of results. Due to the nature of systematic reviews, the possibility of publication bias cannot be completely excluded, as studies with positive findings are more likely to be published than those with negative or inconclusive results.

Although it aligns with the goals of our paper, the literature meeting the inclusion criteria regarding the role of NRAS in MSI-H patients with non-metastatic colon cancer has not been identified. This gap underscores a critical need, as it is desirable for this subpopulation to be included in clinical studies, both in terms of prognosis and prediction. Notably, the limited data available on patients with concurrent NRAS mutations reveal an even more unfavorable prognosis, particularly in patients with metastatic colon tumors, and may be associated with a diminished response to conventional chemotherapy treatments. This impact on chemotherapy response can influence the prognosis and treatment options for patients with non-metastatic colon cancer harboring NRAS mutations, but its role in MSI patients still needs to be defined.

In summary, the negative impact of RAS and BRAF mutations in CRC patients is profound and derives from the MSI status. Understanding the intricate interplay between these mutations and other molecular features is crucial for devising effective therapeutic strategies and improving the overall management of CRC, particularly in the pursuit of personalized and targeted treatment approaches.

Furthermore, our review raises the question of the practical application of MSI assessment as part of the diagnostic routine in the adjuvant setting, alongside RAS and BRAF mutations.

## 5. Conclusions

The literature data consistently show a negative prognostic impact of KRAS and BRAF mutations in MSI patients on long-term outcomes such as RFS and OS, emphasizing the adverse effects of these mutations in metastatic disease rather than in operable cases. The presence of BRAF mutations, in particular, warrants more vigilant clinical monitoring and tailored surveillance strategies due to their association with poorer prognoses. Similarly, the prognostic implications of RAS mutations, including NRAS, necessitate a heightened clinical awareness to tailor treatment plans effectively.

This information gains added relevance with the advent of immunotherapy, which maintains its therapeutic benefits even in patients with BRAF and KRAS mutations. The potential of immunotherapy to offer significant benefits at the onset of metastasis highlights the urgent need for further research to refine treatment options for this specific patient subgroup.

In the pursuit of more personalized antitumor treatments and considering the goal of minimizing adjuvant chemotherapy when the risk-benefit ratio is unfavorable, this review underscores the imperative to further explore the predictive and prognostic roles of BRAF and RAS mutations in patients with MSI. Future studies are essential to better define the benefits of adjuvant therapy in the MSI population, which typically has a highly favorable prognosis, and to assess how the presence of other concurrent genetic mutations impacts survival in the context of new therapeutic possibilities at recurrence. Such research will be crucial for advancing treatment modalities and improving patient outcomes, fostering the development of more effective and personalized therapeutic strategies in oncology.

## Figures and Tables

**Figure 1 diagnostics-14-01001-f001:**
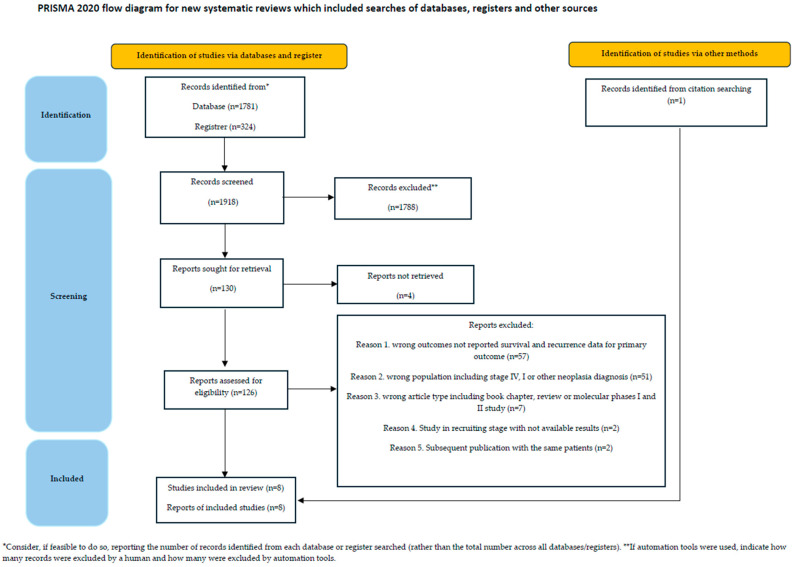
PRISMA 2020 flow-chart.

**Figure 2 diagnostics-14-01001-f002:**
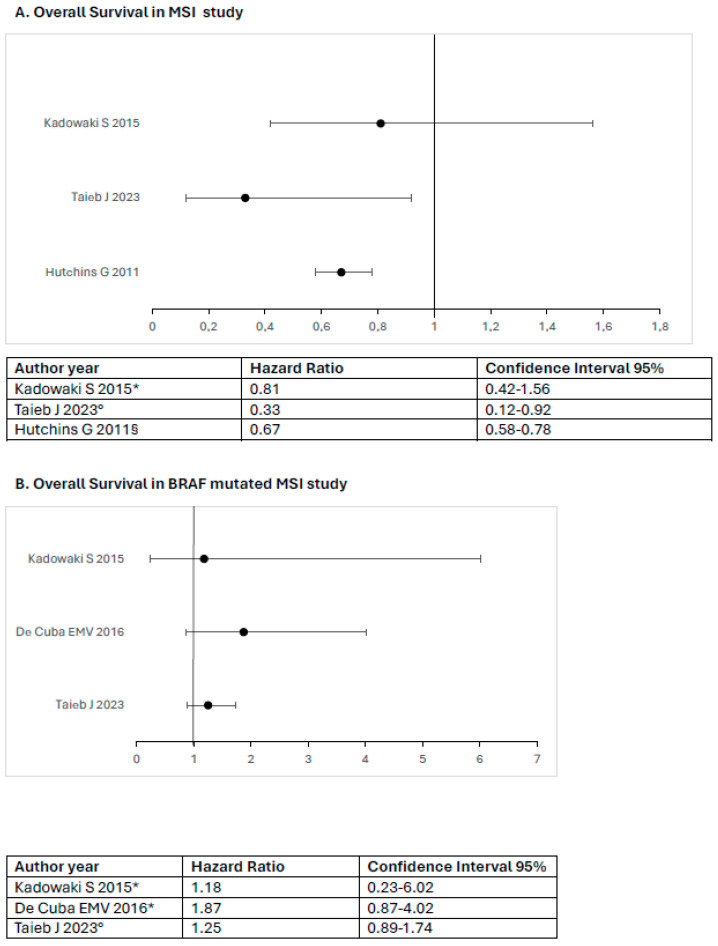
The hazard ratio of survival in the included study. * Observational study. ° Pooled analysis or mixed analysis. § RCTs [36,37,38,41,42].

**Figure 3 diagnostics-14-01001-f003:**
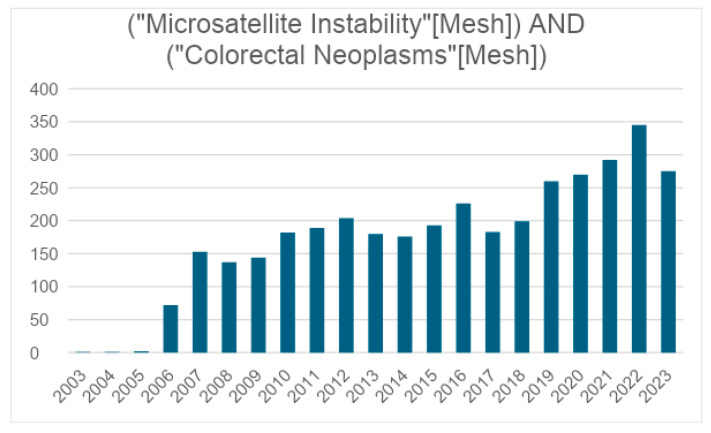
The monitoring of scientific information on results of (“Microsatellite instability”[Mesh] AND (“Colorectal Neoplasm”[Mesh]) (Pubmed).

**Table 1 diagnostics-14-01001-t001:** Characteristics of the study analyzed.

Author, Year	Study Type	Sample *n*	MSI *n* (% Total)	Sex (MSI) (%M)	Stage I MSI (%)	Stage II MSI (%)	Stage III MSI (%)	KRAS MSI (%)	BRAF MSI (%)	Adjuvant Treatment	Primary Outcome of the Study
Hutchins G, 2011 [38]	RCT	1913	218 (11%)	107 (49%)	0 (0%)	205 (94%)	10 (4.6%)	31 (14.2%)	58 (26.6%)	Observation vs. 5FU/LV	Risk of recurrence RR dMMR = 0.53 CI (0.4–0.7)
Taieb J, 2023 [36]	Pooled analysis RCTs	8460	968 (11.4%)	455 (47%)	0 (0%)	0 (0%)	968 (100%)	175 (18%)	393 (40%)	5 FU, FOLFOX, CAPOX, FOLFOX- BEVA, FOLFOX-CET, FOLFIRI CET	TTR, OS, and SAR significantly shorter in patients with mutKRAS and mutBRAF tumors.
Domingo E, 2018 [37]	Mixed analysis *	977	244 (14%)	105 (43%)	0 (0%)	111 (45%)	133 (55%)	36 (15%)	116 (48%)	CAPE, CAPE-BEV, 5FU	mutKRAS, mutBRAF, and TP53, and lower mutation burden were all independently associated with poor prognosis, whereas MSI was not
Maestro M L, 2006 [39]	Obs	351	24 (6.9%)	9 (4.9%)	12 (6.7%)	8 (8.8%)	4 (5.0%)	NA	9 (37.5%)	13 (7.2%)	MSI independent prognostic value (OS)
Batur S, 2016 [40]	Obs	145	28 (19.3%)	8 (28.6%)	NA	NA	11 (40.7%)	NA	7 (25%)	NA	MSI is not related to poor OS; negative wtBRAF status is related to better OS (*p* = 0.048)
De Cuba E M V, 2016 [41]	Obs	143	143 (100%)	62 (43%)	0 (0%)	85 (59%)	58 (41%)	23 (16%)	73 (51%)	36 (25%) CAPE, CAPE-BEV, 5FU	mutKRAS and mutBRAf poorer prognosis in MSI (*p* = 0.04)
Kadowaki S, 2015 [42]	Obs	812	144 (17.7%)	NA	NA	NA	NA	22 (15.5%)	24 (16.6%)	NA	mutKRASand mutBRAF poorer DFS and OS
Nakaji Y, 2017 [43]	Obs	472	44 (9.3%)	21 (47.7)	NA	NA	17 (38.6)	NA	17 (38.6%)	NA	Poorer OS in mutBRAF (*p* = 0.04). In MSI, no OS difference in wtBRAF vs. mutBRAF (*p* = 0.4655)

* The study by Domingo E (QUASAR2 plus Australian), being a combined pooled analysis of an RCT associated with a community-based series, has been excluded from the RCTs. TTR: Time To Recurrence. OS: overall survival. SAR: survival after recurrence. RR: relative risk. Obs: Observational; RCT: Randomized Controlled Trial; NA: Not Available. 5 FU: 5fluorouracil; LV: Leucovorin; CAPE: Capecitabine; BEV: Bevacizumab; CET: Cetuximab.

**Table 2 diagnostics-14-01001-t002:** Survival outcomes in MSI KRAS, NRAS, and BRAF mutation study.

Author, Year	OS MSI	OS MSI mutBRAF	OS MSI mutKRAS	DFS-PFS MSI	DFS-PFS MSI mutBRAF	DFS-PFS MSI mutKRAS	SAR mutBRAF	SAR mutKRAS
Hutchins G, 2011 [38]	NA	NA	NA	NA	RR = 1.32 CI (0.8–2.16) *	NA	NA	NA
Taieb J, 2023 [36]	HR = 0.67 CI (0.58–0.78) vs. MSS °	HR = 1.25 CI (0.89–1.74) °	HR = 1.24 CI (0.84–1.83) °	NA	HR = 1.04 CI (0.75–1.44) °	HR = 1.01 CI (0.69–1.47) °	HR = 1.99 CI (1.30–3.03) °	HR 1.81 CI (1.11–2.93) °
Domingo E, 2018 [37]	NA	NA	NA	HR = 0.90 CI (0.56–1.45) °	HR = 0.55 CI (0.35–0.90) (vs all wild type, MSS) °	HR = 0.28 CI (0.09–0.89) (vs all wild type, MSS) °	NA	NA
Maestro M L. 2006 [39]	HR = 0.33 CI (0.12–0.92) vs. MSI low *	50% (mut) vs. 84% (wt) at 43 months *p* = 1 *	NA	NA	NA	NA	NA	NA
Batur S, 2016 [40]	32 mean months CI (26.7–37.4) *	14.4 ± 7.6 months *p* = 0.001 vs. wtBRAF *	NA	NA	NA	NA	NA	NA
De Cuba E M V, 2016 [41]	Stage II: 82% at 5 years Stage III: 71% at 5 years *	HR = 1.87 CI (0.87–4.02) *	HR = 1.61 CI (0.6–4.33) *	NA	NA	NA	NA	NA
Kadowaki S, 2015 [42]	HR = 0.81 CI (0.42–1.56) °	HR = 1.18 CI (0.23–6.02) °	HR = 1.39 CI (0.33–5.78) °	HR = 0.64 CI (0.35–1.16) °	HR = 2.46 CI (0.49–12.4) °	HR = 1.34 CI (0.34–5.24) °	NA	NA
Nakaji Y, 2017 [43]	*p*-value = 0.4429; HR = 1.423 *	*p*-value = 0.4655; HR = 0.6443 *	NA	*p*-value = 0.2626 HR = 1.57 *	NA	NA	NA	NA

* Univariate analysis. ° Multivariate analysis. NA: Not Available. HR: hazard ratio. CI: 95% Confidence Interval.

## Data Availability

The authors confirm that the data supporting the findings of this study are available within the article and its Appendix A.

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
