# Peer review of "Unraveling the Interplay of KRAS, NRAS, BRAF, and Micro-Satellite Instability in Non-Metastatic Colon Cancer: A Systematic Review"

_diagnostics, 2024, doi:10.3390/diagnostics14101001_

Round 1
Reviewer 1 Report
Comments and Suggestions for Authors
Figure 1 is a very informative flowchart and its content is clear. Maybe just in order to improve the aesthetics of the drawing you can adjust indents inside the blocks. The same is actual for the Table 1.
It would be better to indicate references on the publications in the Tables 1 and 2, column “Author, year”.
Please decipher the abbreviation “RCT” the first time it is mentioned in the text. Maybe it was done but I didn’t notice. The same is actual for other abbreviations which are not very common.
Line 270 “A growing trend in the field of MSI in colon cancer patients has become evident. To illustrate, there was a substantial increase in publications on PubMed from 2006 to 2023.” It is likely a misprint. Maybe you meant trend in the studying of MSI?
Also, there are some minor misprints in the text, for example, in line 273 “Figure 2)” without one of the brackets; words “NA: Not Available HR: Hazard Ratio CI: 95% Confidence Interval” in lines 273-275; just symbol “f” in Table S2, row rayyan-661156027; duplicates of numbers 10 and 47 in the list of references.
Comments on the Quality of English LanguageI think that the text is written in good English, but there are some grammatic errors, and the text can be improved by native speaker, if there is such a possibility.
Author Response
Response to Reviewer 1 Comments
|
||
1. Summary |
|
|
Thank you very much for taking the time to review this manuscript. Please find the detailed responses below and the corresponding revisions/corrections highlighted/in track changes in the re-submitted files.
|
||
3. Point-by-point response to Comments and Suggestions for Authors |
||
Comments 1: Figure 1 is a very informative flowchart and its content is clear. Maybe just in order to improve the aesthetics of the drawing you can adjust indents inside the blocks. The same is actual for the Table 1.
|
||
Response 1: Thank you for pointing this out. We agree with this comment. Therefore, we have adjusted indents inside the blocks of Figure 1 (page 11) and Table 1 (page 12,13).
|
||
Comments 2: It would be better to indicate references on the publications in the Tables 1 and 2, column “Author, year” |
||
Response 2: Agree. We have, accordingly, done the indicate publication in column “Author, year” as requested in Table 1 (page 12,13) and Table 2 (page 13,14).
Comments 2: Please decipher the abbreviation “RCT” the first time it is mentioned in the text. Maybe it was done but I didn’t notice. The same is actual for other abbreviations which are not very common Response 3: Thank you for pointing this out. We agree with this comment. Therefore, we performed the deciphering of RCT (Randomized Controlled Trial) first time in the text at page 4 tab 73. Moreover, we corrected according deciphering in the text the following terms:
- MSI (Microsatellite Instability) Page 3 - tab 54 - DFS (Disease Free Survival) page 4,5 tab 81,82. - RFS (Relapse Free Survival) page 5 – tabs 82,83. - MMR (mismatch repair) page 3 - tab 51. - CRC (Colorectal cancer) we added in the page 2 - tab47 - OS (Overall Survival) we added the decipher in in text Page 5 – tab 84. - pMMR (proficient mismatch repair): page 5– 82. - SAR (survival after recurrence): we added the decipher in text page 6-tab103. - MSS (microsatellite stable): We corrected the decipher at the first time the abbreviation appear in the text at page 5 tab 83.
Comments 4: Line 270 “A growing trend in the field of MSI in colon cancer patients has become evident. To illustrate, there was a substantial increase in publications on PubMed from 2006 to 2023.” It is likely a misprint. Maybe you meant trend in the studying of MSI? Response 4: Agree. We have, accordingly, done the correction of misprint as suggested is now locate in text at page 22 – tab 305 |
||
Comments 5: Also, there are some minor misprints in the text, for example, in line 273 “Figure 2)” without one of the brackets; words “NA: Not Available HR: Hazard Ratio CI: 95% Confidence Interval” in lines 273-275; just symbol “f” in Table S2, row rayyan-661156027; duplicates of numbers 10 and 47 in the list of references. Response 5: Thank you for pointing this out. We agree with this comment. Therefore, we corrected the misprint of (figure 2) as suggested is now locate in text at page – tab and we delete the misprinted word that have been flagged. Moreover, we corrected the right title in Table S2 in row rayyan-661156027 while unfortunately after carefully check we didn’t detect duplicates numbers 10 and 47 in the list of references.
4. Response to Comments on the Quality of English Language |
||
Point 1: I think that the text is written in good English, but there are some grammatic errors, and the text can be improved by native speaker, if there is such a possibility. |
||
|
||
5. Additional clarifications |
||
none |
Reviewer 2 Report
Comments and Suggestions for Authors
Some suggestions are important to improve the weight of this manuscript (review), which are as below;
1. Justify the use of the word ‘role’ in the title. What do you mean by 'current literature' in the title? Please explain. The title of the present research is purely classical. The title should be creative, concisely composed, and relevant to the study’s objectives.
2. Starting your abstract with a word like 'approximately' seems inappropriate generally. Line 13–14: Explain (in detail) with relation to MSI prevalence how it guides the clinician in the choice of potential adjuvant therapy'. Line 15: Why did the authors use the word 'clarify' here? In the abstract, the methodological and result parts are not clearly reflected according to the object, so the author should clarify them. The abstract should be more elaborative. The content presented in this section should make the overall significance and conceptual advance of the present research work clearly accessible to a global reader.
3. In Line 42, why is the author mentioning that MSI is more frequent in stage II (almost 20%) and III (12%) tumors than in stage IV tumors (4%)? Please elaborate on this citation. The national and international status of individual genes related to non-metastatic MSI colon cancer should be mentioned. What was the purpose of the authors for mentioning lines 69–71? Please explain. Line 74, please remove the word 'previous'. The starting lines of the last paragraph should be re-checked again. Authors must ensure that the study's objectives are clearly stated before concluding this section.
4. Methods were written in a recipe-like manner. The authors must subdivide this section into different sub-headings to make this information an interesting segment of this review for global readers. Please also remove some unnecessary details from this section. In lines 147–148, the details for categorized outcomes need more clarification. The stated information should be more focused and well-related to the steps included to complete this work.
5. In some places, authors mentioned that during the initial search, eligibility was assessed. What does this mean? Please improve these statements, and they should be written in a more organized way. I totally reject Figure 1. Please re-draw it in a simpler format, utilizing a meaningful approach to generalize the sequential steps conducted to complete this work. Re-format Table 1 and try to make it more conclusive by eliminating some unwanted variables. Revise the title of Table 2. A univariate or multivariate statistical approach should be indicated with any sign and marked as a footnote. Unfortunately, the meaningful inferences from tables and figures were found to be misleading. I would suggest they relate genes' HR status with published studies through the aid of Forest Plot. See the statement mentioned on lines 202-205. I think the author should re-check the other mentioned findings. Lines 241-242, please justify why the authors mentioned this statement. After reading the observational studies segment, I will suggest they rewrite this section in a more detailed manner. What is the need for mentioning Figure 2? Please justify. I would recommend adding some figures to this work, like a breakdown some information from figures 1 and 2, that can be used in some separate pie charts or other plots, to make this review more meaningful for readers.
6. The discussion section is very weak; the authors repeated many sentences from the methodology and results parts; it's better they explained more about their work in comparison with other studies and results. Please re-check the statement mentioned in lines 332–334. See lines 335–336. Explain the impact of this sentence. Several lines are like a mismatch of information, probably due to over-referencing. The entire section should be broken into some established paragraphs that describe the information on the mentioned topic in a conical manner. The conclusion is very vague; revise this section with future prospects and significant details.
7. Write a strength and limitation section for this work as to how it will benefit the readers. What is the main purpose for global readers to read this review article and utilize its message in their research?
Comments on the Quality of English LanguageMinor editing required.
Author Response
|
|
|
|
|
|
Point-by-point response to Comments and Suggestions for Authors
|
||
Comments 1: Justify the use of the word ‘role’ in the title. What do you mean by 'current literature' in the title? Please explain. The title of the present research is purely classical. The title should be creative, concisely composed, and relevant to the study’s objectives. |
||
Response 1: Thank you sincerely for your thoughtful feedback and for taking the time to review our manuscript. To address the observations about possible confusion with the words "role" and "current literature", we have reconsidered and refined the title of our manuscript. We wholeheartedly acknowledge your suggestions for enhancing the creativity, conciseness, and relevance of our title. We changed the title in “Unraveling the Interplay of KRAS, NRAS, BRAF, and Micro-Satellite Instability in non-metastatic Colon Cancer: A Systematic Review"
|
||
Comments 2: Starting your abstract with a word like 'approximately' seems inappropriate generally. Line 13–14: Explain (in detail) with relation to MSI prevalence how it guides the clinician in the choice of potential adjuvant therapy'. Line 15: Why did the authors use the word 'clarify' here? In the abstract, the methodological and result parts are not clearly reflected according to the object, so the author should clarify them. The abstract should be more elaborative. The content presented in this section should make the overall significance and conceptual advance of the present research work clearly accessible to a global reader. |
||
Response 2: Thank you for pointing this out. We have, accordingly, changed abstract section in order to emphasize this point.
Comments 3: In Line 42, why is the author mentioning that MSI is more frequent in stage II (almost 20%) and III (12%) tumors than in stage IV tumors (4%)? Please elaborate on this citation. The national and international status of individual genes related to non-metastatic MSI colon cancer should be mentioned. What was the purpose of the authors for mentioning lines 69–71? Please explain. Line 74, please remove the word 'previous'. The starting lines of the last paragraph should be re-checked again. Authors must ensure that the study's objectives are clearly stated before concluding this section. Response 3: Thank you for your insightful query regarding the role of MSI. We appreciate the opportunity to clarify this point further. In recent years, immunotherapy has emerged as a transformative approach in cancer treatment, especially for tumors exhibiting high microsatellite instability (MSI-H). The statement, "Immunotherapy has been achieving promising results in the neoadjuvant setting for those patients with the presence of MSI in both monotherapy and combination regimens, confirming the predictive role of MSI even in non-metastatic disease," underscores the significant therapeutic potential and predictive value of MSI status. Recent clinical trials have shown that neoadjuvant immunotherapy, applied before the main treatment, yields substantial benefits in MSI-H non-metastatic colorectal cancers. These benefits include higher response rates and potential for complete pathological responses compared to traditional chemotherapy regimens. MSI retains its predictive importance for the efficacy of immunotherapy in non-metastatic cancer, suggesting that its role extends beyond a simple prognostic marker to a pivotal determinant of therapeutic strategy. We hope this explanation elucidates the significant implications of MSI status for the selection and timing of immunotherapy in non-metastatic colon cancer, reflecting its profound impact on patient outcomes. Regarding your query on the references cited in the Background section, we have carefully revised our citations to ensure accuracy and coherence. Specifically, we have made the following corrections and clarifications: Expression and Mutation Patterns: The reference to the study conducted in the USA, which examines the correlation between mismatch repair protein expression and mutation patterns with MSI status and survival rates in non-metastatic colorectal cancer, is now correctly cited as: Kheirelseid EAH, Miller N, Chang KH, Curran C, Hennessey E, Sheehan M, Kerin MJ. Mismatch repair protein expression in colorectal cancer. J Gastrointest Oncol. 2013;4(4):397-408. [3] International Research on MSI: The systematic review and the broader European research indicating that MSI-high tumors typically exhibit a better prognosis and different responses to chemotherapy, thereby supporting the use of MSI status in clinical trials and treatment planning, are consistently cited as: Popat S, Hubner R, Houlston RS. Systematic review of microsatellite instability and colorectal cancer prognosis. J Clin Oncol. 2005;23(3):609-618. [7] These references correctly reflect our discussion on the importance of adapting treatment protocols based on MSI status and the implications for the use of immunotherapy in MSI-high patients, as highlighted in various international settings. We hope that these clarifications address your concerns satisfactorily. We are committed to ensuring the highest standards of accuracy and reliability in our publication and are grateful for your contributions towards achieving this goal. On line 74, we removed the word "previous." The last paragraph was re-checked as requested, and we have expanded the aim to improve the study’s objectives (page 6, lines 104-106).
Comments 4: Methods were written in a recipe-like manner. The authors must subdivide this section into different sub-headings to make this information an interesting segment of this review for global readers. Please also remove some unnecessary details from this section. In lines 147–148, the details for categorized outcomes need more clarification. The stated information should be more focused and well-related to the steps included to complete this work.
Response 4: Thank you for pointing this out. We have, accordingly, changed the methods section with sub-heading in order to simplify the design and the conduct of the article as suggested. In lines 147-148, we did not include details for categorized outcomes because our work is a systematic review that does not involve statistical analysis; we merely reported the results of the included studies. Therefore, since our objective is focused on survival and disease progression outcomes, we have not included details related to categorical outcomes in that section
Comments 5: In some places, authors mentioned that during the initial search, eligibility was assessed. What does this mean? Please improve these statements, and they should be written in a more organized way. I totally reject Figure 1. Please re-draw it in a simpler format, utilizing a meaningful approach to generalize the sequential steps conducted to complete this work. Re-format Table 1 and try to make it more conclusive by eliminating some unwanted variables. Revise the title of Table 2. A univariate or multivariate statistical approach should be indicated with any sign and marked as a footnote. Unfortunately, the meaningful inferences from tables and figures were found to be misleading. I would suggest they relate genes' HR status with published studies through the aid of Forest Plot. See the statement mentioned on lines 202-205. I think the author should re-check the other mentioned findings. Lines 241-242, please justify why the authors mentioned this statement. After reading the observational studies segment, I will suggest they rewrite this section in a more detailed manner. What is the need for mentioning Figure 2? Please justify. I would recommend adding some figures to this work, like a breakdown some information from figures 1 and 2, that can be used in some separate pie charts or other plots, to make this review more meaningful for readers. Response 5: Dear Reviewer, regarding the term "eligibility," it is defined in our manuscript as the evaluation of articles assessed for inclusion in the systematic review according to the inclusion and exclusion criteria. We agree with your comment on Figure 1. Therefore, we have reformatted Figure 1 in a simpler format as per the PRISMA 2020 statement (page 11). On page 13, line 203, we have changed the title of Table 2. The univariate or multivariate approach is now indicated with a mark in the table legend as a footnote. We refer to what was previously Figure 2 (now Figure 3 in this version) to emphasize the relevance of the topic in the context of colon cancer and to justify the importance of performing a systematic review on this topic. Finally, to make the results more immediate and intelligible and to clarify the statements, we have added a new Figure 2 with forest plots of the survival results from the included studies (pages 19, 20, 21).
Comments 6: The discussion section is very weak; the authors repeated many sentences from the methodology and results parts; it's better they explained more about their work in comparison with other studies and results. Please re-check the statement mentioned in lines 332–334. See lines 335–336. Explain the impact of this sentence. Several lines are like a mismatch of information, probably due to over-referencing. The entire section should be broken into some established paragraphs that describe the information on the mentioned topic in a conical manner. The conclusion is very vague; revise this section with future prospects and significant details. Response 6: Thank you for your insightful feedback regarding the structure and content of the "Discussion" section in our manuscript. We have taken your recommendations into careful consideration and have revised this section accordingly to enhance clarity and coherence.
We have reorganized the "Discussion" into distinct, thematic paragraphs to ensure a logical flow and to address the complex interactions of the topics covered:
MSI Research and Clinical Implications: This paragraph discusses the growing trend in MSI research and the clinical implications of MSI as a prognostic tool in CRC, highlighting the increased focus over the years and its significance in therapeutic decision-making. BRAF and RAS Mutations: Comparative Analysis with Other Studies: We delve into the roles of BRAF and RAS mutations in CRC pathogenesis, comparing our findings with existing studies and discussing the specific challenges posed by these mutations in MSI contexts. Strength and Limitations: as you suggested in Comments 7. Conclusion: We have thoroughly revised and detailed this section to address your concerns, ensuring it now encompasses a clear articulation of future prospects and significant details relevant to the study’s findings. Regarding the role of NRAS in our discussion, we have clarified that the absence of data highlighted by our review specifically pertains to the impact that NRAS mutations may have in MSI patients.
Comments 7: Write a strength and limitation section for this work as to how it will benefit the readers. What is the main purpose for global readers to read this review article and utilize its message in their research? Response 7: Dear reviewer, we carefully considered your recommendations and will incorporate the necessary adjustments as suggested in the revised version of our paper. Here the strength and limitation section: This systematic review synthesizes a comprehensive range of studies examining the prognostic significance of NRAS, KRAS, and BRAF mutations within non-metastatic MSI colon cancer. A major strength of this review is its extensive coverage of diverse genetic profiles, which provides a nuanced understanding of how these mutations influence patient outcomes across different populations. The review utilizes rigorous methodological standards to ensure that the findings are reliable and replicable, enhancing the utility of the conclusions drawn for clinical practice. Despite its strengths, this review has several limitations. The studies included may have varied in design and scope, potentially introducing heterogeneity that could affect the interpretation of results. Due to the nature of systematic reviews, the possibility of publication bias cannot be completely excluded, as studies with positive findings are more likely to be published than those with negative or inconclusive results. We have included this section prior to discussing other significant limitation already documented in the manuscript (“Although it aligns with the goals of our paper, literature meeting the inclusion criteria regarding the role of NRAS in patients with non-metastatic colon cancer has not been identified. This gap underscores a critical need, as it is desirable for this subpopulation to be included in clinical studies, both in terms of prognosis and prediction.”)
The primary purpose of this review is to consolidate current knowledge on the prognostic implications of key genetic mutations in non-metastatic MSI colon cancer, thereby guiding more effective and personalized treatment strategies. It aims to foster a deeper understanding of the molecular dynamics at play, which is crucial for advancing treatment modalities and improving patient outcomes on a global scale. We have therefore revised the conclusions, detailing and more clearly defining the purpose as previously requested in comments number 6. |
||
|
Round 2
Reviewer 2 Report
Comments and Suggestions for Authors
Accept in present form.